# Adding chemically selective subtraction to multi-material 3D additive manufacturing

David Gräfe[1,2], Andreas Wickberg[3], Markus Michael Zieger[1], Martin Wegener[3,4], Eva Blasco[1] & Christopher Barner-Kowollik[1,2]

Existing photoresists for 3D laser lithography that can be removed after development in a subtractive manner typically suffer from harsh cleavage conditions. Here, we report chemoselectively cleavable photoresists for 3D laser lithography based on silane crosslinkers, allowing the targeted degradation of 3D printed microstructures under mild conditions. Three bifunctional silane crosslinkers carrying various substitutions on the silicon atom are synthesized. The photoresists are prepared by mixing these silane crosslinkers with pentaerythritol triacrylate and a two-photon photoinitiator. The presence of pentaerythritol triacrylate significantly enhances the direct laser written structures with regard to resolution, while the microstructures remain cleavable. For the targeted cleavage of the fabricated 3D microstructures, simply a methanol solution including inorganic salts is required, highlighting the mild cleavage conditions. Critically, the photoresists can be cleaved selectively, which enables the sequential degradation of direct laser written structures and allows for subtractive manufacturing at the micro- and nanoscale.

[1] Macromolecular Architectures, Institute for Technical Chemistry and Polymer Chemistry, Karlsruhe Institute of Technology (KIT), Engesserstr. 18, 76128 Karlsruhe, Germany. [2] School of Chemistry, Physics and Mechanical Engineering, Queensland University of Technology (QUT), 2 George Street, Brisbane, QLD 4000, Australia. [3] Institute of Applied Physics, KIT, Wolfgang-Gaede-Str. 1, 76131 Karlsruhe, Germany. [4] Institute of Nanotechnology, KIT, Hermann-von-Helmholtz-Platz 1, 76344 Eggenstein-Leopoldshafen, Germany. Correspondence and requests for materials should be addressed to M.W. (email: martin.wegener@kit.edu) or to E.B. (email: eva.blasco@kit.edu) or to C.B-K. (email: christopher.barnerkowollik@qut.edu.au)

The direct structuring of soft matter in three dimensions via direct laser writing (DLW), also referred to as 3D laser lithography or 3D laser printing, has revolutionized the field of micro-optics[1–4]. This light-based micro- and nanoscale printing method is based on a multi-photon polymerization of a photoresist employing a femtosecond laser, in which two (and occasionally more) photons are absorbed simultaneously. Due to the nonlinearity of the multi-photon process, the chemical reaction occurs exclusively within the focal spot of the laser and allows for locally defined crosslinking. Thus, DLW is capable of fabricating complex 3D structures on the submicron length scale. This high resolution—in some instances sub-diffraction resolution by exploiting STED principles[5–8]—is particularly attractive for applications where sophisticated structures with high precision are required, including optical metamaterials, biomedicine, microfluidics, and microelectronic[1,9–12]. To date, there exists no other 3D manufacturing approach on the submicron length scale that would be even remotely as versatile.

An important research field for current and future applications of DLW is to extend the range of available functional photoresists[4]. Classically, 3D microstructures obtained via DLW are irreversibly crosslinked into a permanent form. For many applications, however, it is important that the 3D structure consists of a material, which is removable or replaceable at a later stage. Such an approach, which has also been termed subtractive manufacturing, is particularly interesting for applications where natural damage results in a short lifetime and material parts need to be replaced. Furthermore, direct laser written 3D structures have been used as templates for complex yet well-controlled architectures of inorganic materials or for the design of flying features in specific structures[13]. However, existing resists suffer from incomplete template removal and distortion of cavities due to harsh cleavage conditions. Thus, progress in these research areas has been limited by the provision of appropriate degradable photoresists.

It is noteworthy that the availability of photoresists for 3D laser lithography that can be cleaved on demand is still in its infancy and only recently we have introduced a first photoresist leading to microstructures that can be degraded upon a defined chemical trigger[14]. The next frontier in cleavable photoresist design is the ability of the resulting structures to be removed by orthogonal chemical triggers, allowing the targeted removal of specific material elements within a structure. Ideally, the number of selectively removable parts in a 3D structure should be as large as possible enabling the fabrication of previously unknown classes of multifunctional and complex materials. Here, we introduce a class composed of three such photoresists capable of 3D laser lithography, based on chemospecifically addressable silane linkers. We take inspiration from linkers employed in the realm of degradable biomaterials[15], and demonstrate that well-defined 3D microstructures become accessible. Critically, we show that completely selective cleavage can be achieved, with the individual cleavage conditions not affecting the fabricated structures written with disparate material properties. As a proof of principle, we report the controlled degradation of three individual microstructures on the same substrate in a sequential manner.

## Results

**Designing cleavable photoresists for 3D laser lithography.** Silyl ethers are commonly utilized as protective groups for alcohols in organic chemistry[16,17]. The stability of silyl ethers, i.e., the ease of cleavage, depends on the substituent on the silicon atom. In general, sterically bulky substituents on the silicon atom decrease the rate of cleavage. As a result, triethylsilyl ethers are more stable than trimethylsilyl ethers, yet less stable than triisopropylsilyl ethers towards acid or base hydrolysis. This difference in silyl ether stability allows selective deprotection of individual hydroxyl groups in the same molecule, which is commonly employed in the organic synthesis of natural products. In analogy to established strategies for degradable soft materials, we prepared crosslinkers with labile linkages based on silyl ethers that undergo cleavage under mild yet specific conditions. Notably, we have synthesized acrylamide-based crosslinkers to introduce additional physical crosslinks, i.e., hydrogen bonds. It has been shown that networks composed of amide bonds exhibit much higher stiffness and stability compared to their ester analogs[18,19]. This aspect is particularly beneficial for true 3D nano- and microstructures having freely suspended segments. In addition to physical crosslinks, the degradation of DLW structures benefits from the higher hydrophilic character of amide bonds[20]. It was observed that degradation of DLW structures is slow due to the high crosslinking density[14]. Increasing the hydrophilicity allows for greater swelling of written structures in polar solvents as here employed, associated with greater accessibility of the cleavable bonds. Figure 1 illustrates the synthesis of crosslinkers with a labile silyl ether linkage. Specifically, three crosslinkers containing methyl (MSEA), ethyl (ESEA) or isopropyl (ISEA) substituents on the silicon atom were prepared. In a typical procedure, the chlorosilane with the substituents of interest was added to a solution containing hydroxyethyl acrylamide and triethylamine. After purification of the crosslinkers by column chromatography, structural confirmation was obtained by electrospray ionization mass spectrometry (ESI MS), $^1$H and $^{13}$C nuclear magnetic resonance (NMR) spectroscopy (for details see Supplementary Figs. 1–9). In all cases, the proton resonances found in the NMR spectra are in agreement with the corresponding structure and integration values matched with the number of protons. In addition, molecular ion peaks obtained by high resolution ESI MS are in agreement with the calculated data. An observation of considerable significance is that all three crosslinkers are liquids, allowing photoresist formulations without the addition of solvents. This simplicity is highly important for further usage, including industrial applications.

In order to assess the performance of the synthesized crosslinkers in DLW, every silane-based bisacrylamide was mixed with the two-photon photoinitiator Irgacure 369 (2-benzyl-2-(dimethylamino)-1-[4-(morpholinyl) phenyl)]-1-butanone) and

**Fig. 1** Synthesis of acrylamide-based crosslinkers with labile silyl ether linkage in a one-step synthesis. Crosslinkers with methyl (MSEA), ethyl (ESEA) or isopropyl (ISEA) as substituents on the silicon atom were prepared

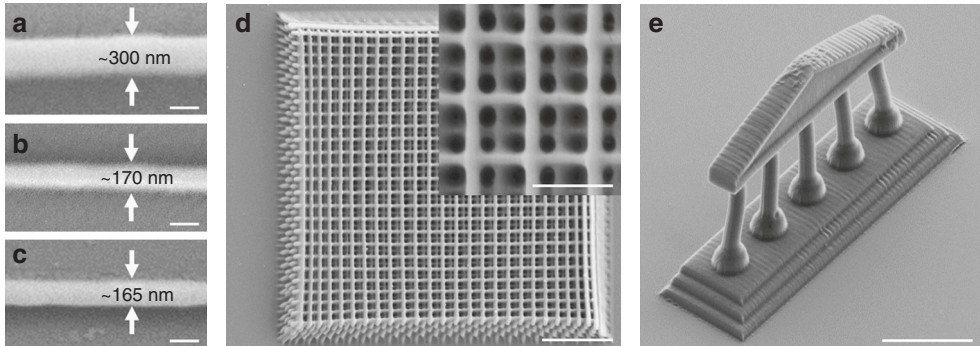

**Fig. 2** SEM images of polymer lines and 3D structures written on a glass substrate. Polymer line composed of **a** 100 mol% ISEA, **b** 97.5 mol% ISEA and 2.5 mol% PETA, and **c** 100 mol% PETA (scale bars = 200 nm). **d** Woodpile structure composed of 97.5 mol% ISEA and 2.5 mol% PETA (scale bars = 1 and 4 µm, respectively). **e** Fragment of a Greek temple with multiple columns holding up an entablature composed of 97.5 mol% ESEA and 2.5 mol% PETA (scale bar = 10 µm)

employed for the fabrication of line patterns to determine the minimum feature size (i.e., the linewidth). It is advantageous for a DLW photoresist to support a high spatial resolution and small feature sizes. All experimental DLW was carried out with a commercially available 3D DLW lithography system (Photonics Professional by Nanoscribe GmbH) using an oil-immersion configuration. Line patterns were fabricated with a writing speed of 50 µm s$^{-1}$ and a laser power of 5 mW. We found that these writing conditions are close to the photopolymerization threshold. As a useful reference point for our study, we additionally performed the same experiment with commercially available pentaerythritol triacrylate (PETA). PETA is a commonly used multi-functional monomer for DLW[4,7,21,22]. Figure 2 displays representative lines fabricated either with 100 mol% ISEA or 100 mol% PETA that were written on a glass substrate. The linewidth achievable with ISEA is close to 300 nm, while a narrower linewidth was achieved for PETA, close to 165 nm. It has been shown that the conversion degree in DLW, thus the crosslinking density, depends on the laser power and writing speed[23,24]. Because all polymer lines were fabricated with the same writing conditions, we assume that an increase in the number of polymerizable groups per molecule translates to a higher crosslinking density of 3D structures allowing for thinner lines. In order to evaluate whether this assumption is correct, we added PETA to the photoresist to increase the crosslinking density. It is important to note that we kept the PETA amount at 2.5 mol% to retain the cleavable properties of the direct laser written structure. We observed that when the PETA concentration is above 2.5 mol%, non-cleavable 3D structures were fabricated (for more details see Supplementary Figs. 14 and 15). We found a remarkable improvement in the minimum linewidth when PETA was added. The presence of 2.5 mol% of PETA reduced the linewidth to 170 nm, which is close to the value of the reference material composed of 100 mol% PETA while retaining the desired cleavage properties. It is notable that a slightly wider linewidth was observed for MSEA (linewidth 230 nm) compared to ESEA and ISEA (linewidth for both 170 nm). We assume that this is attributed to the different viscosities of the three photoresists. While MSEA is a low-viscous liquid, ESEA and ISEA are both highly viscous oils. The higher viscosity of ESEA and ISEA reduces the lateral flow preventing fabricated structures from blurring and allows for thinner linewidths[25,26].

**Fabrication of complex 3D structures**. Once the writing conditions were optimized and allowed writing high-resolution lines, we

turned our attention to true 3D structures. Woodpiles were selected as a model structure because it is a standard benchmark structure in the field of DLW due to the complexity of its topology[1–4]. All woodpiles were fabricated at identical writing conditions as for the line pattern. Figure 2d depicts an SEM image of a woodpile structure made of a photoresist composed of 97.5 mol% ISEA and 2.5 mol% PETA. Specifically, a woodpile containing 12 layers with a footprint of 20 µm × 20 µm and a rod spacing of $a = 800$ nm was fabricated. This woodpile showed a moderate level of shrinkage (percentage of shrinkage of 10%, Table S2) with well-separated layers and well-aligned rods without apparent defects. Critical for powerful DLW are high writing speeds, enabling rapid manufacturing of 3D microstructures. Therefore, we assessed our photoresists by fabricating 3D model structures. For example, a fragment of a Greek temple with multiple columns holding up an entablature at a writing speed of 1 mm s$^{-1}$ and a laser power of 12.5 mW using another photoresist composed of 97.5 mol% ESEA and 2.5 mol% PETA was successfully fabricated. This step further highlights the powerful writing properties of our photoresists (refer to Fig. 2e). It should be noted that for the fabrication of complex 3D structures, the use of a pure silane crosslinker resulted in non-detailed and deformed structures and the addition of PETA was necessary. Consequently, we only used our advantageous photoresist mixture composed of 97.5 mol% silane crosslinker and 2.5 mol% PETA for all further studies. For other examples of polymer lines and 3D microstructures at various writing conditions see Supplementary Figs. 10–12.

**Orthogonality studies**. It is critical for current and future DLW applications that single elements of a direct laser written structure can be removed chemospecifically under mild conditions without affecting the remaining other parts of the 3D structure. This enables, for example, removal of heavily stressed parts within a 3D microstructure or even complete 3D microstructures when used as templates. Ideally, a set of photoresists allows for multiple removals in an orthogonal manner. Through a comprehensive review of the literature on silyl ether protecting groups, three reagent systems were identified as excellent candidates for the selective and efficient cleavage of our direct laser written structures depending on the employed silane crosslinker.[16,17] Specifically, we found that the use of sodium hydrogen carbonate, potassium carbonate and potassium fluoride in methanol serves our needs for selective cleavage.

In order to assess the orthogonality of our cleavable photoresists, we fabricated direct laser written microstructures

composed of MSEA, ESEA, ISEA or PETA and subjected them to three conditions: (1) a saturated solution of $NaHCO_3$ in MeOH at 50 °C, (2) a saturated solution of $K_2CO_3$ in MeOH at room temperature (RT), and (3) a saturated solution of KF in MeOH at RT. The cleavage of the 3D microstructures was visualized by scanning electron microscopy (SEM) and optical microscopy (for details see Supplementary Figs. 16–20). A summary of the stability screening is presented in Table 1. It should be emphasized that all degradation experiments were carried out within a certain timeframe to ensure selective cleavage of individual structures. Degradation studies using $NaHCO_3$ in MeOH at 50 °C showed complete removal of MSEA-based structures within 20 min, while structures composed of ESEA, ISEA or PETA remained entirely unaffected. It is notable that microstructures composed of MSEA were stable to $NaHCO_3$ in MeOH at RT. When subjected to a saturated solution of $K_2CO_3$ in MeOH, 3D structures made of MSEA and ESEA were degraded. In contrast, direct laser written structures based on ISEA or PETA proved to be stable under similar conditions. In addition, the stability of all structures towards fluoride-ions by immersing them in a solution of KF was screened. 3D structures composed of MSEA and ISEA underwent degradation, whereas structures containing ESEA and PETA remained stable. The stability of microstructures based on ESEA is particularly surprising because fluoride-ions readily attack silicon and cleave the silyl ether bond. Overall, our orthogonality studies demonstrate that MSEA can be cleaved, while ESEA and ISEA remain intact. In addition, the two photoresists ESEA and ISEA are orthogonal to each other in terms of degradation. Both photoresists can be selectively cleaved one over another, and vice versa.

**Table 1 Orthogonality studies of direct laser written structures composed of MSEA, ESEA, ISEA or PETA under three different conditions**

| Conditions | MSEA | ESEA | ISEA | PETA |
|---|---|---|---|---|
| $NaHCO_3$, 50 °C, 20 min | ✗ [a] | ✓ [b] | ✓ | ✓ |
| $K_2CO_3$, RT, 1 h | ✗ | ✗ | ✓ | ✓ |
| KF, RT, 1 h | ✗ | ✓ | ✗ | ✓ |

All degradation experiments were conducted with saturated solutions in MeOH
[a] 3D structure cleaved
[b] 3D structure stable

**Selective cleavage of DLW microstructures.** Once the unique selectively of our cleavable photoresists was proven, we fabricated MSEA-, ESEA-, ISEA-, and PETA-based model structures on a single glass substrate, and degraded them in a consecutive manner as shown in Fig. 3. For clarity, we decided to use only individual structures, as the cleavage process can be readily monitored by optical microscopy. Nevertheless, the described resist system could also be combined into multi-material architectures[14,27]. First, time-lapse images using optical microscopy were taken at different times in order to monitor the degradation process (see Supplementary Fig. 21). In addition, SEM images were recorded after each cleavage step to obtain further detailed information about the remaining microstructures. Additional SEM images were also taken when the structures were partially degraded. Based on our previous orthogonality studies revealing that MSEA structures can be cleaved selectively, while ESEA, ISEA, and PETA structures remain stable (refer to Table 1), the microstructures were initially treated with $NaHCO_3$ in MeOH at 50 °C. SEM images clearly evidence that the MSEA-based Eiffel tower degraded partially after 10 min, and disappeared completely after 20 min. As expected, 3D microstructures composed of ESEA, ISEA, and PETA were stable at these conditions. Subsequently, the sample was immersed in a solution of $K_2CO_3$ resulting in the degradation of the Aztec temple composed of ESEA within 1 h. SEM images revealed a blurry Aztec temple after 30 min. The resulting sample containing two microstructures was finally subjected to KF in MeOH. At these conditions, SEM showed partial cleavage of the bridge based on ISEA after 30 min. Complete removal was observed within 1 h. Importantly, our reference microstructure, i.e., the Asian temple, made of PETA, remained unaffected and was stable during all three cleavage steps. These results highlight the efficient and rapid cleavage under mild conditions without affecting the conventional photoresist based on acrylates (i.e., PETA).

**Discussion**

In summary, we have introduced a class of cleavable photoresists for 3D laser lithography based on labile silane crosslinkers, allowing the targeted and subsequent degradation of three different direct laser written 3D microstructures. Specifically, three silane crosslinkers with methyl, ethyl, or isopropyl as substituents on the silicon atom were prepared using a facile and rapid one-step synthesis. We observed that addition of 2.5 mol% PETA significantly enhanced the direct laser written structures with regard to resolution, while the 3D microstructures remained cleavable. For the cleavage of the 3D microstructures, we only employed inorganic salts, i.e., $NaHCO_3$, $K_2CO_3$, and KF, in

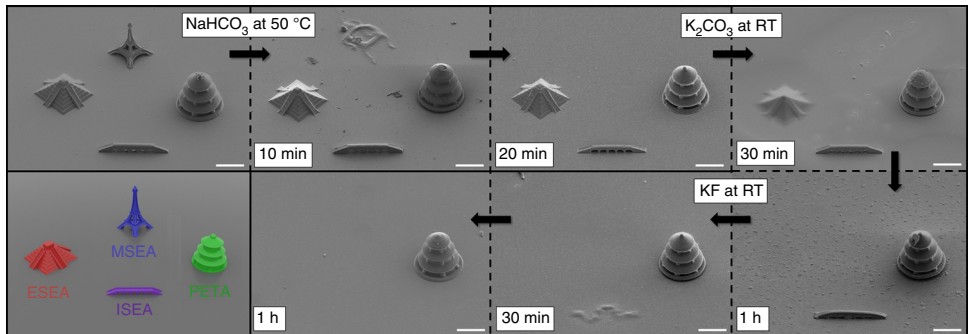

**Fig. 3** SEM images showing the selective cleavage of MSEA-, ESEA-, and ISEA-based 3D microstructures. The glass substrate was sequentially immersed in $NaHCO_3$ at 50 °C, then, in $K_2CO_3$ at RT and finally, in KF at RT (scale bars = 20 μm). (Bottom left) Schematic representation of a glass substrate containing four model structures composed of MSEA (blue Eiffel tower), ESEA (red Aztec pyramid), ISEA (purple bridge) and PETA (green Asian temple)

methanol. The degradation process was visualized by scanning electron and optical microscopy, showing rapid and complete removal independent of the used photoresist. Our degradation studies demonstrate the unique chemospecificity of our pioneered cleavable photoresist. MSEA-based structures can be cleaved, while ESEA- and ISEA-based structures remain intact. Furthermore, ESEA- and ISEA-based structures are orthogonal to each other. Both photoresists can be selectively cleaved one over another, and vice versa. In order to emphasize the chemospecificity of our set of photoresists, we degraded MSEA-, ESEA-, and ISEA-based 3D microstructures on a single glass substrate in three consecutive steps. Critically, our reference 3D microstructure made of PETA remained unaffected during these cleavage steps highlighting the mild cleavage conditions. Due to the mild, efficient and selective nature of the cleavage process, we submit that the class of photoresists presented here hold large potential and will allow for the fabrication of a variety of complex and multifunctional 3D nano- and microstructures that are presently inaccessible using current state of the art photoresists and/ or subtractive manufacturing methodologies. In addition, we anticipate the fabrication of multi-materials using our photoresist system can be expanded to other 3D printing technologies, including stereolithography and digital micromirror device-based projection printing.

## Methods

**Materials**. All chemicals were used without further purification. Dichlorodimethylsilane (98%, abcr), dichloromethane (p.a., VWR), diethyldichlorosilane (97%, abcr), diisopropyldichlorosilane (97%, abcr), dimethylformamide (p.a., VWR), dimethylsufoxide-$d_6$ (99.8% D, euriso-top), ethyl acetate (p.a., VWR), Irgacure 369 (Ciba Inc.), hydroxyethyl acrylamide (97%, Sigma-Aldrich), pentaerythritol triacrylate (technical grade, Sigma-Aldrich), sodium bicarbonate (Sigma-Aldrich), 3-(trimethoxysilyl)propyl methacrylate (98%, Sigma-Aldrich), potassium fluoride (Sigma-Aldrich), potassium carbonate (Sigma-Aldrich), triethylamine (99%, Acros), MeOH (anhydrous, 99.8%, Sigma-Aldrich).

**Nuclear magnetic resonance spectroscopy**. Proton nuclear magnetic resonance ($^1$H NMR) spectra were recorded on a Bruker AM 400 (400 MHz) spectrometer. Chemical shifts are expressed in parts per million (ppm) and calibrated on characteristic solvent signals as internal standards. Carbon nuclear magnetic resonance ($^{13}$C NMR) spectra were recorded on a Bruker AM 400 (101 MHz) spectrometer.

**Mass spectrometry**. High-resolution mass spectra were conducted via electron spray ionization mass spectrometry utilizing a Q Exactive (Orbitrap) mass spectrometer (Thermo Fisher Scientific, San Jose, CA, USA) equipped with a HESI II probe. Calibration of the instrument was carried out in the m z$^{-1}$ range of 74–1822 using calibration solutions from Thermo Scientific. The FT resolution was set to 140,000 employing 3 microscans during an acquisition time between 2 and 3 min measuring. The spray voltage was set to 4.7 kV and a dimensionless sheath gas flow of 5 was applied. The capillary temperature was set to 320 °C and the S-lens value was set to 62.0. The injection was performed with a flow rate of 5 μL min$^{-1}$.

**Scanning electron microscopy**. SEM images were recorded using a Zeiss Supra 55VP. All samples were sputter coated with a 10 nm gold layer.

**Optical microscopy**. Time-lapse microscopy was carried out on a Zeiss Axiophot microscope.

**Synthesis**. For the characterization data ($^1$H NMR, $^{13}$C NMR, and ESI MS) for all molecules see Supplementary Figs. 1–9.

***N,N*′-(((Dimethylsilanediyl)bis(oxy))bis(ethane-2,1-diyl))diacrylamide (MSEA)**. To a 100 mL Schlenk round-bottom flask equipped with a magnetic stirrer hydroxyethyl acrylamide (HEA) (8.30 g, 31.0 mmol), triethylamine (NEt$_3$) (13 mL) and 40 mL dichloromethane (DCM) were added. After stirring for 10 min, dichlorodimethylsilane (2.50 g, 7.75 mmol) in 7 mL DCM was added dropwise and allowed to stir overnight for 12 h at ambient temperature. DCM was removed under reduced pressure and the crude product was purified by column chromatography (Al$_2$O$_3$, DCM: acetonitrile: NEt$_3$ – 92: 4: 4). After the residual solvent was removed under reduced pressure, MSEA was obtained as a colorless liquid. Yield: 1.2 g (54%).

$^1$H NMR (DMSO-$d_6$, 400 MHz): δ (ppm) = 0.07 (s, 6 H), 3.24 (q, $J$ = 5.9 Hz, 4 H), 3.65 (t, $J$ = 5.9 Hz, 4 H), 5.58 (dd, $J$ = 10, 3 Hz, 2 H), 6.07 (dd, $J$ = 16.5, 2.8 Hz, 2 H), 6.24 (dd, $J$ = 17, 10.4 Hz, 2H), 8.15 (m, 2H)
$^{13}$C NMR (DMSO-$d_6$, 101 MHz): δ (ppm) = −3.18 (CH$_3$), 40.94 (CH$_2$), 60.59 (CH$_2$), 125.06 (CH$_2$), 131.68 (CH), 164.73 (CO)
MS (m z$^{-1}$) calculated for C$_{12}$H$_{22}$N$_2$O$_4$Si, [M]$^+$ = 286.1343, [M + Na]$^+$ = 309.1241; found [M + Na]$^+$ = 309.1241

***N,N*′-(((Diethylsilanediyl)bis(oxy))bis(ethane-2,1-diyl))diacrylamide (ESEA)**. To a 100 mL Schlenk round-bottom flask equipped with a magnetic stirrer HEA (4.01 g, 34.83 mmol), NEt$_3$ (9.5 mL) and 40 mL DCM were added. After stirring for 10 min, dichlorodiethylsilane (2.64 g, 16.79 mmol) in 7 mL DCM was added dropwise and allowed to stir overnight for 12 h at ambient temperature. DCM was removed under reduced pressure and the crude product was purified by column chromatography (silica gel, hexane: ethyl acetate: NEt$_3$ – 30: 69: 1). After the residual solvent was removed under reduced pressure, ESEA was obtained as a colorless oil. Yield: 3.27 g (62%).

$^1$H NMR (DMSO-$d_6$, 400 MHz): δ (ppm) = 0.56 (q, $J$ = 7.8 Hz, 4H), 0.9 (t, $J$ = 8.0 Hz, 6H), 3.25 (q, $J$ = 6.1 Hz, 4H), 3.67 (t, $J$ = 6.0 Hz, 4H), 5.57 (dd, $J$ = 10, 2.6 Hz, 2H), 6.07 (dd, $J$ = 17, 2. Hz, 2H), 6.24 (dd, $J$ = 17, 10.2 Hz, 2H), 8.14 (m, 2H)
$^{13}$C NMR (DMSO-$d_6$, 101 MHz): δ (ppm) = 3.21 (CH$_3$), 6.28 (CH$_2$), 41.03 (CH$_2$), 60.69 (CH$_2$), 125.07 (CH$_2$), 131.68 (CH), 164.76 (CO)
MS (m z$^{-1}$) calculated for C$_{12}$H$_{22}$N$_2$O$_4$Si, [M]$^+$ = 314.1656, [M + Na]$^+$ = 337.1554; found [M + Na]$^+$ = 337.1449

***N,N*′-(((Diisopropylsilanediyl)bis(oxy))bis(ethane-2,1-diyl))diacrylamide (ISEA)**. To a 100 mL Schlenk round-bottom flask equipped with a magnetic stirrer HEA (3.33 g, 29 mmol), NEt$_3$ (9 mL) and 50 mL DCM were added. After stirring for 10 min, dichlorodiisopropylsilane (1.54 g, 8.28 mmol) in 7 mL DCM was added dropwise and allowed to stir overnight for 12 h at ambient temperature. DCM was removed by rotary evaporation under reduced pressure and the product was isolated by column chromatography (silica gel, hexane: ethyl acetate: NEt$_3$ – 30: 69: 1). After the residual solvent was removed under reduced pressure, ISEA was obtained as a colorless oil. Yield: 1.8 g (65%).

$^1$H NMR (DMSO-$d_6$, 400 MHz): δ (ppm) = 0.98 (m, 14H), 3.27 (q, $J$ = 6.2 Hz, 4H), 3.72 (t, $J$ = 6.2 Hz, 4H), 5.57 (dd, $J$ = 10, 2 Hz, 2H), 6.07 (dd, $J$ = 17, 2.5 Hz, 2H), 6.24 (dd, $J$ = 17.3, 10 Hz, 2H), 8.13 (m, 2H)
$^{13}$C NMR (DMSO-$d_6$, 101 MHz): δ (ppm) = 11.36 (CH), 17.11 (CH$_3$), 41.06 (CH$_2$), 61.02 (CH$_2$), 125.04 (CH$_2$), 131.68 (CH), 164.79 (CO)
MS (m z$^{-1}$) calculated for C$_{12}$H$_{22}$N$_2$O$_4$Si, [M]$^+$ = 342.1969, [M + Na]$^+$ = 365.1867; found [M + Na]$^+$ = 365.1864

**Photoresists without PETA**. In a typical procedure, 250 mmol of the silane crosslinker of interest was mixed with 1.5 mmol of Irgacure 369 and subsequently stirred and ultrasonicated for 1 h before conducting DLW experiments.

**Photoresists with PETA**. In a typical procedure, 243.8 mmol of the silane crosslinker of interest was mixed with 6.2 mmol of PETA and 1.5 mmol of Irgacure 369 and subsequently stirred and ultrasonicated for 1 h before conducting DLW experiments.

**Silanization of glass substrates**. All glass substrates were cleaned with acetone, 2-propanol, ionized water and ultrasonificated for 15 min in acetone. After activating the substrates for 20 min employing air plasma, the glass substrates were immersed in a solution of 3-(trimethoxysilyl)propyl methacrylate in toluene (0.5 mM). Subsequently, the substrates were ultrasonicated in toluene (5 min) and acetone (5 min) to remove physisorbed silane.

**Direct Laser Writing**. The commercially available DLW setup Photonic Professional GT (Nanoscribe GmbH, Germany) was used. The instrument was equipped with a high-numerical aperture (NA = 1.4, ×63) oil immersion objective lens. A mode-locked and frequency-doubled Er-doped fiber laser was employed, emitting femtosecond pulses at 780 nm center wavelength. All employed glass substrates were silanized. Line patterns and woodpiles were fabricated with a writing speed of 50 μm s$^{-1}$ and a laser power of 5 mW. 3D microstructures were fabricated with writing speeds ranging from 1 mm s$^{-1}$ to 10 mm s$^{-1}$ and a laser power close to the damage threshold. For additional information about the writing conditions see Supplementary Figs. 10–12. All structures were developed by immersing the glass substrates in methanol and acetone for 15 min.

**Data availability**. Data supporting the findings of this study are available within the article and its Supplementary Information file, and from the corresponding authors upon reasonable request.

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

## Acknowledgements

D.G. acknowledges the German Research Council (DFG) for funding his post-doctoral studies. C.B.-K. acknowledges the Australian Research Council (ARC) for funding in the context of a Laureate Fellowship enabling his photochemical research program as well as the Queensland University of Technology (QUT) for continued key support. We acknowledge support by the Helmholtz program Science and Technology of Nanosystems (STN) and the Karlsruhe School of Optics & Photonics (KSOP). Further, we acknowledge support by the DFG and the Open Access Publishing Fund of KIT. Open access publication was further enabled by the platform for Manufacturing with Advanced Materials of the Institute for Future Environments (QUT). M.Z. is grateful for a PhD scholarship and the support from the Evangelisches Studienwerk e.V. Villigst.

## Author contributions

D.G., E.B. and C.B.-K. conceived and initiated the study. D.G. synthesized the materials. D.G. and M.Z. performed the DLW experiments. D.G. carried out the degradation experiments. D.G. and A.W. performed the SEM analyses. M.W., E.B. and C.B.-K. motivated and supervised the research program. D.G., E.B. and C.B.-K. wrote a first draft of the manuscript. All authors discussed the results and worked on the manuscript.

## Additional information

**Competing interests:** The authors declare no competing interests.

