## [Peer Review File · Nature Communications]

Reviewers' comments:

Reviewer #1 (Remarks to the Author):

The major claims of the paper:

The major claims of the paper are of significant scientific novelty and practical importance. Definitely it will be of interest for direct laser writing 3D lithography, advanced materials science and 3D printing (in general) communities.

The experimental section is convincing and the extensive Supplementary parts are supporting the claims. However, not all or not all of them are fully validated. And this raises my major concerns, which if properly answered can secure the manuscript for the publication, or alternatively, as a Reviewer I do not see the manuscript to be of such high impact as Nature Communications, but rather Scientific Reports or similar Journal. Both Major and Minor issues are expressed below.

Major questions:

1. In the Abstract and within the main text there is no laser parameters or the dimensions of the structures introduced, which actually is very important as it would tell whether the proposed method is limited to just femtosecond laser 3D micro-/nano-lithography or can be extended to a much wider applications in DLW based UV lithography and projection UV lithography. It also should be discussed in the Summary, however now it is missing.

2. DLW 3D lithography employing ultrafast lasers is known to be of unmatched flexibility (also stated in this manuscript). And it can be used for so called 4D printing where besides 3 dimensional coordinates the 4th one is the material or its properties (degree of polymerization/modification). For instance, recently it was shown by Rekstyte et al in "Microactuation and sensing using reversible deformations of laser-written polymeric structures, Nanotechnology 28, 124001 (2017)", that hybrid (or composite) structures can be made using DLW 3D lithography and used as programmable materials. Why Authors of this manuscript did not show multi-material 3D structures and their selective removal? This would definitely justify the high impact of the chosen journal.

3. Next, the "erasing speed" or cleavage rate of the structures must be dependent on their volume and especially surface-to-volume ratio. Additionally, it is known that the material properties (physical and chemical) depend on their degree of cross-linking (which is mentioned as being denser in the case of mixture with PETA). This can be evaluated using Raman micro-spectroscopy, thus it would enable to determine the "erasing speed" relation to degree of cross-linking (which depends on exposure intensity/dose) and optimize it for specific 3D structures to increase/decrease erasing orthogonality. See for instance: Zukauskas et al in "Tuning the refractive index in 3D direct laser writing lithography: towards GRIN microoptics, Laser Photon. Rev. 9(6), 706-712 (2015)" and Li Jia Jiang et al in "Two-photon polymerization: investigation of chemical and mechanical properties of resins using Raman microspectroscopy, OPTICS LETTERS 39(10) 3034, 2014".

4. Lastly, it should be minor but it is major, since >50% of the references are of Authors' own works. For instance, while introducing STED-lithography the 3 citations only of their own work is given [5-7], even though STED-technique is not directly related to the current report. Possibility to structure non-photosensitized resins while using femtosecond pulses is of adequate importance, for instance: Mechanisms of three-dimensional structuring of photo-polymers by tightly focussed femtosecond laser pulses, Opt Express. 2010 May 10; 18(10):10209-21. doi: 10.1364/OE.18.010209.

And among applications a bunch of 6 references [8-13] is provided, again only covering Authors' own work. Even though Authors keep their research work focus to < 100 μm structures (or similar) due to their limited Nanoscribe setup, there are diverse reports showing the DLW 3D lithography can be successfully applied to large scale production of scaffolds suitable for in vivo studies (Maciulaitis et al in 2015 Biofabrication 7, 015015) or highly efficient parallel fabrication inside microfluidic chip (Bing Xu et al in Scientific Reports 6, 19989 (2016)).

Minor:

1. Woodpile structure is a Photonic crystal if it has corresponding spectral/spatial performance, if it is not validated – should be called adequately just woodpile structure.
2. Figure 2: I recommend to mark that in the figure that the PETA concentration is 2.5 mol%, not just ISEA+PETA as it looks of equal or proportional parts.
3. Is this a Full Greek temple in Fig 2. (E)? I assume its just a fragment, thus so it should be named.

Conclusion:

In case the above expressed questions can be properly addressed, the manuscript can become orbicular for wider audience and be acceptable for the publication in the selected Journal. Otherwise it is interesting and original work, but not yet it its best shape of presentation, thus not reaching the high level of the Nature Communications.

Reviewer #2 (Remarks to the Author):

General comments: The authors have put together a well-written and highly interesting research paper entitled “Adding Chemically Selective Subtraction to Multi-Material 3D Additive Manufacturing”. The presented silane-based labile crosslinkers are of interest for a wider community of researchers dealing with degradable materials. The study is concise and the presented experiments and results support the formulated findings and claims. The paper is also written for a wider audience and fun to read. I strongly recommend the publication of this paper. However, a few questions remained and should be addressed within the manuscript.

-) The silane-based crosslinkers have been designed bearing acrylamide functionality for reasons mentioned in the manuscript (e.g. higher hydrophilicity to enhance hydrolytic degradation). Hence, it would have been more consistent to use a multifunctional acrylamide instead of PETA as reference.

-) It is clear that with highly crosslinking photoresists (incl. PETA) the minimum linewidth can be reduced. However, what is the explanation for the inferior performance of MSEA compared to ESEA and ISEA concerning minimal line width? All three components are difunctional and similar reactivity would be expected. The difference in performance should be discussed.

-) When looking at Figure 3, there seems to be slight degradation for the Aztec temple in sodium carbonate and also slight degradation of the bridge in potassium carbonate. This observed degradation is in disagreement with full chemical orthogonality and I would expect the structures to fully disappear with longer chemical treatment? I would assume that full chemical orthogonality is not achieved, but rather chemical orthogonality via “optimized timing” of the chemical treatment, which should be mentioned as a crucial factor. In my opinion stating this clearly in the main text and conclusion is important for the reader.

-) I am also missing a detailed chemical explanation for the presented chemical orthogonalities. The structures of the three silane-based crosslinkers only differ with the alkyl substituents on the Si-atom. Steric hinderance for sure is a factor yielding different hydrolytic stability and should be mentioned. Are there other explanations, especially for the higher stability for ESEA over ISEA in KF???

-) Have there been attempts to synthesize tri- or tetrafunctional crosslinkers with labile silyl ether linkages? Given the availability of the respective starting materials the synthesis should be straight forward. This could enable the 3D structuring of pure silane-based photoresist formulations.

Some minor comments:

-) Conclusion: "Our degradation studies demonstrates" is grammatically incorrect
-) Page 9: Synthesis. Experimental details and characterization data (1H NMR, 13C NMR, etc.) for all molecules can be found in the Supplementary Information. I suggest to be more specific here and list the characterization data displayed in the SI instead of writing "etc."
-) SI Page 4 section 2.1: "Al3O2" instead of "Al3Ox2"
-) SI Page 15: The picture quality of Figure S14 is not very good. Without descriptive explanations the figure cannot be clearly interpreted.
-) SI Page 16 section 6.2: Supplementary Supplementary

Reviewer #3 (Remarks to the Author):

This paper describes strategies for fabricating 3D structures with multiphoton fabrication in which specific elements can be removed subsequently. The authors have created a set of materials in which individual materials can be removed with different chemistries. This research is quite clever and of great interest, and I fully support its publication in Nature Communications. I think that it will be of high impact and will stimulate a significant amount of new science and applications. I am quite excited about this work. However, a number of issues should be addressed before this manuscript is ready for publication.

First and foremost, the chemistry demonstrated here is not orthogonal, as is made clear by Table 1. A chemical system is either orthogonal or it isn't, there is no such thing as "partial orthogonality" or "multi-orthogonality." Please describe what you have achieved properly, and do not use the term "orthogonal" where it is not warranted. The true state of affairs does not make this work any less important or impressive.

This work demonstrates the dissolution of individual structures. To achieve the full impact of this technology, it would be desirable to dissolve individual portions of single structures. Have the authors been able to achieve this goal? If not, why not? Please address this in the paper.

In the abstract, I am not clear on what the first sentence is intended to imply. Photoremoval of structures created with multiphoton polymerization has been demonstrated (Light: Science & Applications volume 1, page e6 (2012) and references therein). Surely Ormocers can be removed with HF. I think that this sentence should be either removed or justified/qualified.

The work in the paper is unnecessarily oversold at points. For instance, if the authors are not "breaking new ground," there would be no point to the paper at all. Please tone down or remove such statements.

The citations in the paper are overwhelmingly to the work of the authors. This group has done superb work in this field over the years, but so have many other groups. Please give credit where credit is due.

What is "the usual level of shrinkage?" Please define. The shrinkage in Figure 2E seems rather extreme, is this really usual?

Reviewer #1

The major claims of the paper are of significant scientific novelty and practical importance. Definitely it will be of interest for direct laser writing 3D lithography, advanced materials science and 3D printing (in general) communities. The experimental section is convincing and the extensive Supplementary parts are supporting the claims. However, not all or not all of them are fully validated. And this raises my major concerns, which if properly answered can secure the manuscript for the publication, or alternatively, as a Reviewer I do not see the manuscript to be of such high impact as Nature Communications, but rather Scientific Reports or similar Journal. Both Major and Minor issues are expressed below.

We thank the reviewer for her/his supportive comments and have made a very serious effort to accommodate all the noted suggestions into the manuscript.

Major questions:

1. In the Abstract and within the main text there is no laser parameters or the dimensions of the structures introduced, which actually is very important as it would tell whether the proposed method is limited to just femtosecond laser 3D micro-/nano-lithography or can be extended to a much wider applications in DLW based UV lithography and projection UV lithography. It also should be discussed in the Summary, however now it is missing.

We thank the reviewer very much for the comment and we agree that laser parameters are missing in the main text. For this reason, we have included at each fabricated structure detailed information regarding the writing conditions. For example, we have added in the main text (page 4):

“Line patterns were fabricated with a writing speed of 50 $\mu\text{m/s}$ and a laser power of 5 mW.”

In addition, we included the following short section in the method part (page 9):

“Line patterns and woodpiles were fabricated with a writing speed of 50 $\mu\text{m/s}$ and a laser power of 5 mW. 3D microstructures were fabricated with writing speeds ranging from 1 mm/s to 10 mm/s and a laser power close to the damage threshold. Additional information about the writing conditions can be found in the Supplementary Information.”.

With regard to the dimensions of the fabricated structures, we kindly note that we already indicated in the main text the dimensions of some the microstructures (e.g. page 5 “a woodpile containing 12

layers with a footprint of $20\ \mu\text{m} \times 20\ \mu\text{m}$ and a rod spacing of $a = 800\ \text{nm}$ was fabricated”). In addition, all SEM images contain a scale bar, which allows the reader to judge the structures’ dimension.

Since we are aiming to introduce cleavable photoresists, which allow for subtractive manufacturing, we decided to focus on the main properties of the photoresist system without giving specific information about the laser parameter in the abstract.

Concerning a wider application in the field of DLW, we agree with the reviewer that this point can additionally emphasised. Although our study mainly focuses on 3D printing at the microscale using 3D laser lithography, we submit that the acrylamide-based monomers described can be used in UV lithography and projection UV lithography. Thus, the following sentences have been incorporated in the conclusion (page 9):

“In addition, we anticipate the fabrication of multi-materials using our photoresist system can be expanded to other 3D printing technologies including stereolithography and digital micromirror device-based projection printing.”

- 2. DLW 3D lithography employing ultrafast lasers is known to be of unmatched flexibility (also stated in this manuscript). And it can be used for so called 4D printing where besides 3 dimensional coordinates the 4th one is the material or its properties (degree of polymerization/modification). For instance, recently it was shown by Rekstyte et al in “Microactuation and sensing using reversible deformations of laser-written polymeric structures, Nanotechnology 28, 124001 (2017)”, that hybrid (or composite) structures can be made using DLW 3D lithography and used as programmable materials. Why Authors of this manuscript did not show multi-material 3D structures and their selective removal? This would definitely justify the high impact of the chosen journal.**

The reviewer raises an interesting point, yet the primary aim of our study is to focus on the selective cleavage of a 3D printed material, while other material elements remain unaffected. This can be shown either on individual structures – as we did – or by a hybrid structure, where single elements are fabricated of different materials. However, these two experimental approaches are both multi-materials that are on the same substrate and differ only in the spatial distance of the different materials. Consequently, a cleavage experiment would result in the same outcome. We do not believe that the results of this manuscript would significantly improve when performing the same experiments again with a hybrid structures. Critically, however, there are several other advantages of using individual structures: The cleavage process of our proof-of-concept cannot be monitored with optical microscopy because it requires structures in the micron range and only allows a top view. In addition, hybrid structures have been reported by several groups as the reviewer notes and their fabrication is in

principle possible with every photoresist. Nevertheless, we agree with the reviewer that this point needs to be addressed in the manuscript. Thus, we have added in the main text (page 7):

“For clarity, we decided to use only individual structures because the cleavage process can easily be monitored by optical microscopy. Nevertheless, the described resist system could also be combined into multi-material architectures.²⁵”

3. Next, the “erasing speed” or cleavage rate of the structures must be dependent on their volume and especially surface-to-volume ratio. Additionally, it is known that the material properties (physical and chemical) depend on their degree of cross-linking (which is mentioned as being denser in the case of mixture with PETA). This can be evaluated using Raman micro-spectroscopy, thus it would enable to determine the “erasing speed” relation to degree of cross-linking (which depends on exposure intensity/dose) and optimize it for specific 3D structures to increase/decrease erasing orthogonality. See for instance: Zukauskas et al in “Tuning the refractive index in 3D direct laser writing lithography: towards GRIN microoptics, *Laser Photon. Rev.* 9(6), 706-712 (2015)” and Li Jia Jiang et al in “Two-photon polymerization: investigation of chemical and mechanical properties of resins using Raman microspectroscopy, *OPTICS LETTERS* 39(10) 3034, 2014”.

We agree with the reviewer that the cleavage rate depends on the crosslinking density as well as the surface-to-volume ratio. It is obvious that reducing the crosslinking density would result in a faster cleavage process. For our cleavage experiments, however, we exclusively fabricated 3D microstructures with a laser power close to the damage threshold with writing speeds between 1 and 3 mm/s. Consequently, there is no question that the structures are strongly crosslinked, as otherwise no free form 3D printing with each photoresists would be possible. Assuming equal crosslinking density for all fabricated structures, optimized conditions have been shown by the orthogonal cleavage of individual structures as shown in Figure 3. We kindly note that in the context of the line width discussion, we indeed explore an indirect effect on the crosslinking density. We have thus added the following section to the manuscript:

“The line width achievable with ISEA is close to 300 nm, while a narrower line width was observed for PETA, close to 165 nm. It has been shown that the conversion degree in DLW, thus the crosslinking density, depends on the laser power and writing speed.^{23,24} Because all polymer lines were fabricated with the same writing conditions, we assume that an increase in the number of polymerizable groups per molecule translates to a higher crosslinking density of 3D structures allowing for thinner lines. In order to evaluate whether this assumption is correct, we added PETA to the photoresist to increase the crosslinking density. It is important to note that we kept the PETA amount at 2.5 mol% to retain the cleavable properties of the direct laser written structure. We observed that when the PETA

concentration is above 2.5 mol%, non-cleavable 3D structures were fabricated (refer to the Supplementary Information for more details, Figs. S14-15).”

In addition, we have cited the noted studies in the appropriate context.

- 4. Lastly, it should be minor but it is major, since >50% of the references are of Authors' own works. For instance, while introducing STED-lithography the 3 citations only of their own work is given [5-7], even though STED-technique is not directly related to the current report. Possibility to structure non-photosensitized resins while using femtosecond pulses is of adequate importance, for instance: Mechanisms of three-dimensional structuring of photo-polymers by tightly focussed femtosecond laser pulses, Opt Express. 2010 May 10;18(10):10209-21. doi: 10.1364/OE.18.010209. And among applications a bunch of 6 references [8-13] is provided, again only covering Authors' own work. Even though Authors keep their research work focus to < 100 µm structures (or similar) due to their limited Nanoscribe setup, there are diverse reports showing the DLW 3D lithography can be successfully applied to large scale production of scaffolds suitable for in vivo studies (Maciulaitis et al in 2015 Biofabrication 7, 015015) or highly efficient parallel fabrication inside microfluidic chip (Bing Xu et al in Scientific Reports 6, 19989 (2016)).**

We thank the reviewer for pointing this issue out with which we agree. We have carefully revised our manuscript with regard to self-citations and adapted it according to the reviewer's suggestions. Specifically, we have included studies of Seet *et al.* (DOI:10.1002/adma.200401527), Xu *et al.* (DOI: 10.1038/srep19989) and Radke *et al.* (DOI:10.1002/adma.201100543), among others. Overall, there are now 27 references in the revised manuscript, of which we have authored four. Thus, the self-citation rate is at 14 %. We note that this percentage is low, considering that we have worked in this field for many years already. We hope that the change is in line with the reviewer's thoughts.

Minor:

- 1. Woodpile structure is a Photonic crystal if it has corresponding spectral/spatial performance, if it is not validated – should be called adequately just woodpile structure.**

We thank the reviewer for the valuable suggestion. We have corrected this misleading wording.

2. **Figure 2: I recommend to mark that in the figure that the PETA concentration is 2.5 mol%, not just ISEA+PETA as it looks of equal or proportional parts.**

We thank the reviewer for the recommendation. We have removed all unnecessary marks in the figure to avoid any misunderstandings. The exact composition of the photoresist is explained in the figure caption.

3. **Is this a Full Greek temple in Fig 2. (E)? I assume it's just a fragment, thus so it should be named.**

We thank the reviewer for his careful reading of our manuscript. We have corrected the main text and the caption of the figure according to the reviewer's suggestions.

Conclusion:

In case the above expressed questions can be properly addressed, the manuscript can become orbicular for wider audience and be acceptable for the publication in the selected Journal. Otherwise it is interesting and original work, but not yet it its best shape of presentation, thus not reaching the high level of the Nature Communications.

Because reviewer #1 noted some concerns about the impact of our study and the shape of presentation, we would like to point out important arguments for inclusion of our manuscript in *Nature Communication*:

- It constitutes the first report about a set of orthogonal cleavable photoresists capable of 3D laser lithography
- It presents a 4D material by adding subtractive properties to an additive 3D printed material
- It holds high potential also for other research fields, including photopolymer network formation, macroscopic 3D printing and degradable soft matter materials.

We note that our self-assessment of the novelty and impact of the study is in excellent agreement with the recommendations of the other two reviewers.

Reviewer #2

The authors have put together a well-written and highly interesting research paper entitled “Adding Chemically Selective Subtraction to Multi-Material 3D Additive Manufacturing”. The presented silane-based labile crosslinkers are of interest for a wider community of researchers dealing with degradable materials. The study is concise and the presented experiments and results support the formulated findings and claims. The paper is also written for a wider audience and fun to read. I strongly recommend the publication of this paper. However, a few questions remained and should be addressed within the manuscript.

We thank the reviewer for her/his supportive comments and have made a strong effort to accommodate all the noted improvements to the manuscript.

- 1. The silane-based crosslinkers have been designed bearing acrylamide functionality for reasons mentioned in the manuscript (e.g. higher hydrophilicity to enhance hydrolytic degradation). Hence, it would have been more consistent to use a multifunctional acrylamide instead of PETA as reference.**

We agree that a multifunctional acrylamide crosslinker could potentially be better suited as a reference for our study than PETA. To the best of our knowledge, the only commercially available multifunctional acrylamide crosslinker is *N,N'*-methylenebis(acrylamide). Despite the fact that *N,N'*-methylenebis(acrylamide) is little used in the field of 3D laser lithography, mainly in the realm of hydrogels, it is a solid and would require the use of a solvent for the photoresist formulation. Thus, it does not constitute a suitable reference because our cleavable structures are written without solvent in their formulation. In addition, usage of solvent should be avoided in 3D laser lithography as it reduces the overall stability of true 3D structures. Therefore, the commonly used acrylate photoresist PETA was chosen as a reference for this work. PETA's writing properties are well-known in the field and can readily be used as a benchmark to assess the performance of new photoresists.

- 2. It is clear that with highly crosslinking photoresists (incl. PETA) the minimum linewidth can be reduced. However, what is the explanation for the inferior performance of MSEA compared to ESEA and ISEA concerning minimal line width? All three components are bifunctional and similar reactivity would be expected. The difference in performance should be discussed.**

We thank the reviewer for her/his note and agree that this difference in performance needs further explanation in the manuscript. Thus, we have added a short section comparing the minimal line widths of our three photoresists (page 5):

“It is notable that a slightly wider linewidth was observed for MSEA (linewidth 230 nm) compared to ESEA and ISEA (linewidth for both 170 nm). We assume that this is attributed to the different viscosities of the three photoresists. While MSEA is a low-viscous liquid, ESEA and ISEA are both highly viscous oils. The higher viscosity of ESEA and ISEA reduces the lateral flow preventing fabricated structures from blurring and allows for thinner linewidths.^{22,23}”

- 3. When looking at Figure 3, there seems to be slight degradation for the Aztec temple in sodium carbonate and also slight degradation of the bridge in potassium carbonate. This observed degradation is in disagreement with full chemical orthogonality and I would expect the structures to fully disappear with longer chemical treatment? I would assume that full chemical orthogonality is not achieved, but rather chemical orthogonality via “optimized timing” of the chemical treatment, which should be mentioned as a crucial factor. In my opinion stating this clearly in the main text and conclusion is important for the reader.**

We thank the reviewer for pointing this detail out and fully agree that degradation entails a time axis. Long-term exposure of labile materials such as ours to unfavourable conditions can induce undesired cleavage. However, we did not observe degradation or blurring of our structures in the employed timeframe. SEM images demonstrate that the remaining structures had been unaffected. Longer chemical treatment was also not necessary because the target structure was already removed. In addition, selective cleavage of silyl ether protecting groups depending on the substituent has been reported in numerous studies. It is notable that all degradation experiments were initially performed with macroscopic films over several hours or even days. These experiments showed similar results as presented here. To point out that time is an important parameter for the usage of cleavable photoresists, we have added the temporal dimension to our stability screening in Table 1 and included the following statement into the main text (page 6):

“It should be emphasized that all degradation experiments were carried out within a certain timeframe to ensure selective cleavage of individual structures.”

4. **I am also missing a detailed chemical explanation for the presented chemical orthogonalities. The structures of the three silane-based crosslinkers only differ with the alkyl substituents on the Si-atom. Steric hinderance for sure is a factor yielding different hydrolytic stability and should be mentioned. Are there other explanations, especially for the higher stability for ESEA over ISEA in KF?**

We agree with the reviewer that a detailed explanation of the silyl ether chemistry is missing in the main text of the manuscript. Consequently, we have added a paragraph addressing the stability of silyl ethers at the beginning of the results section to adequately introduce the reader to the chemistry of silyl ethers. We have added the following (page 3) text segment:

“Silyl ethers are commonly utilized protective groups for alcohols in organic chemistry.^{16,17} The stability of silyl ethers, *i.e.*, the ease of cleavage, depends on the substituent on the silicon atom. In general, sterically bulky substituents on the silicon atom decrease the rate of cleavage. As a result, triethylsilyl ethers are more stable than trimethylsilyl ethers, yet less stable than triisopropylsilyl ethers towards acid or base hydrolysis. This difference in silyl ether stability allows selective deprotection of individual hydroxyl groups in the same molecule, which is commonly employed in the organic synthesis of natural products.”

Concerning the stability of ESEA and ISEA in KF, the result was also surprising to us, since one expects a similar stability as for trifunctional silyl ether protecting groups. However, the stability of bifunctional silyl ether is rather unexplored and to the best of our knowledge not investigated in detail in the literature. Currently, we cannot provide an in-depth explanation for this result.

5. **Have there been attempts to synthesize tri- or tetrafunctional crosslinkers with labile silyl ether linkages? Given the availability of the respective starting materials the synthesis should be straight forward. This could enable the 3D structuring of pure silane-based photoresist formulations.**

In fact, attempts have also been made to synthesize trifunctional crosslinkers with labile silyl ether linkages. However, it has been found that the synthesis of such molecules is not straightforward due to the high reactivity of chlorosilanes. For an industrially relevant application, however, it is beneficial when the synthesis is rapid and facile, allowing also non-experts the preparation of the resists. This is especially true for physicists or biologists, who use 3D laser lithography to fabricate structures for their individual needs (metamaterials, cell scaffolds, and others). In addition, we demonstrated that photoresists composed of a bifunctional silane crosslinker and PETA are capable of fabricating 3D microstructures with equal resolution limits as the reference material PETA.

Some minor comments:

- 6. Conclusion: “Our degradation studies demonstrates” is grammatically incorrect.**

We thank the reviewer for the hint. We have corrected this grammatical mistake.

- 7. Page 9: Synthesis. Experimental details and characterization data (1H NMR, 13C NMR, etc.) for all molecules can be found in the Supplementary Information. I suggest to be more specific here and list the characterization data displayed in the SI instead of writing “etc.”**

We have changed the terminology to “¹H NMR, ¹³C NMR, and ESI MS” to be more specific about the employed characterization methods.

- 8. SI Page 4 section 2.1: “Al3O2” instead of “Al3Ox2”**

We thank the reviewer for her/his carefully reading. We have replaced “Al₃Ox₂” with “Al₃O₂”.

- 9. SI Page 15: The picture quality of Figure S14 is not very good. Without descriptive explanations the figure cannot be clearly interpreted.**

We agree that picture quality of Figure S14 is poor. For this reason, we have replaced the low-quality image with a higher resolution one.

- 10. SI Page 16 section 6.2: Supplementary Supplementary**

We thank the reviewer for the mistake in our manuscript. We have deleted the repeated word Supplementary.

Reviewer #3

This paper describes strategies for fabricating 3D structures with multiphoton fabrication in which specific elements can be removed subsequently. The authors have created a set of materials in which individual materials can be removed with different chemistries. This research is quite clever and of great interest, and I fully support its publication in Nature Communications. I think that it will be of high impact and will stimulate a significant amount of new science and applications. I am quite excited about this work. However, a number of issues should be addressed before this manuscript is ready for publication.

We thank the reviewer for her/his supportive comments and have made a strong effort to accommodate all the noted improvements to the manuscript.

- 1. First and foremost, the chemistry demonstrated here is not orthogonal, as is made clear by Table 1. A chemical system is either orthogonal or it isn't, there is no such thing as "partial orthogonality" or "multi-orthogonality." Please describe what you have achieved properly, and do not use the term "orthogonal" where it is not warranted. The true state of affairs does not make this work any less important or impressive.**

We would like to thank the reviewer for pointing out this important issue. We have selected more general terms (*i.e.*, 'selective' and 'chemoselective') and replaced misleading terms like "partial orthogonal" or "multi-orthogonality". We hope that the change is in line with the reviewer's thoughts.

- 2. This work demonstrates the dissolution of individual structures. To achieve the full impact of this technology, it would be desirable to dissolve individual portions of single structures. Have the authors been able to achieve this goal? If not, why not? Please address this in the paper.**

As reviewer #1 also raised this point, we refer to our answer in the section above. Alternatively, here our detailed answer to the fabrication of hybrid structures can be found below, quoting from our reply to reviewer #1:

The reviewer raises an interesting point, yet the primary aim of our study is to focus on the selective cleavage of a 3D printed material, while other material elements remain unaffected. This can be shown either on individual structures – as we did – or by a hybrid structure, where single elements are fabricated of different materials. However, these two experimental approaches are both multi-materials that are on the same substrate and differ only in the spatial distance of the different materials. Consequently, a cleavage experiment would result in the same outcome. We do not believe that the

results of this manuscript would significantly improve when performing the same experiments again with a hybrid structures. Critically, however, there are several other advantages of using individual structures: The cleavage process of our proof-of-concept cannot be monitored with optical microscopy because it requires structures in the micron range and only allows a top view. In addition, hybrid structures have been reported by several groups as the reviewer notes and their fabrication is in principle possible with every photoresist. Nevertheless, we agree with the reviewer that this point needs to be addressed in the manuscript. Thus, we have added in the main text (page 7):

“For clarity, we decided to use only individual structures, as the cleavage process can be readily monitored by optical microscopy. Nevertheless, the described resist system could also be combined into multi-material architectures.²⁵”

- 3. In the abstract, I am not clear on what the first sentence is intended to imply. Photoremoval of structures created with multiphoton polymerization has been demonstrated (Light: Science & Applications volume 1, page e6 (2012) and references therein). SurelyOrmocers can be removed with HF. I think that this sentence should be either removed or justified/qualified.**

We agree with the reviewer that the first sentence is misleading and needs revision. As mentioned by the reviewer, there are several photoresists for DLW that can be degraded after development in a subtractive manner. However, it should be noted that they always require very harsh conditions. To clarify our point of view that current cleavable photoresists have several disadvantages, we have replaced the first sentence in the abstract “3D printing on the micrometer scale is classically irreversible“ with “Existing photoresists for 3D laser lithography (Direct Laser Writing (DLW)) that can be removed after development in a subtractive manner typically suffer from harsh cleavage conditions, such as calcination, oxygen-plasma etching, or etching with hydrofluoric acid.”

- 4. The work in the paper is unnecessarily oversold at points. For instance, if the authors are not “breaking new ground,” there would be no point to the paper at all. Please tone down or remove such statements.**

We have carefully revised our manuscript and removed the noted statements. Specifically, we have exchanged the term “we break new ground” with “introducing” or “opens the door” with “allows for”.

- 5. The citations in the paper are overwhelmingly to the work of the authors. This group has done superb work in this field over the years, but so have many other groups. Please give credit where credit is due.**

We thank the reviewer for pointing out this issue with which we agree. We have carefully revised our manuscript with regard to self-citations. Specifically, we have included studies of Seet *et al.* (DOI:10.1002/adma.200401527), Xu *et al.* (DOI: 10.1038/srep19989) and Radke *et al.*(DOI:10.1002/adma.201100543), among others. We hope that the change is in line with the reviewer's thoughts.

6. What is “the usual level of shrinkage?” Please define. The shrinkage in Figure 2E seems rather extreme, is this really usual?

We thank the reviewer for the remark and agree that the term “usual level of shrinkage” is inaccurate. Thus, we have determined the percentage of shrinkage according to literature reports and added this value in the main text. The new text now reads (page 5):

“This woodpile showed a moderate level of shrinkage (percentage of shrinkage of 10 %, Table S2) with well separated layers and aligned rods without apparent defects.”

In addition, we introduced a new section to the Supplementary Information, addressing the shrinkage of each photoresists (Table S2).

REVIEWERS' COMMENTS:

Reviewer #1 (Remarks to the Author):

The Authors have addressed my raised concerns and the manuscript became suitable for publishing in the selected Journal as is.

Reviewer #2 (Remarks to the Author):

The authors have clearly addressed the major concerns, which were brought forward by the reviewers. In my opinion the revision was conducted carefully and the overall manuscript is of high quality. My recommendation is to publish this manuscript in Nature Communications as is.

Reviewer #3 (Remarks to the Author):

The authors have improved the manuscript substantially, and have addressed the majority of my concerns. The only thing that could be improved, as both another reviewer and I brought up, is the demonstration of the removal of parts of a single structure. The authors argue that it must work, given what they have already shown. Although I largely agree with this viewpoint, I also feel that most people who would be interested in this technology would care about exactly such structures. One could imagine that issues such as interdiffusion of different materials and mass transport issues could indeed make it more tricky to develop multimaterial structures selectively. I am therefore sorry that the authors have decided not to address this issue, but do not feel that this omission, although it lowers the impact of the paper somewhat, should keep this work out of Nature Communications.